# Modelling Heterogeneity and Super Spreaders of the COVID-19 Spread through Malaysian Networks

**Fatimah Abdul Razak \*** and **Zamira Hasanah Zamzuri**

Department of Mathematical Sciences, Faculty of Science and Technology, Universiti Kebangsaan Malaysia, Bangi 43600, Selangor, Malaysia; zamira@ukm.edu.my
* Correspondence: fatima84@ukm.edu.my

**Abstract:** Malaysia is multi-ethnic and diverse country. Heterogeneity, in terms of population interactions, is ingrained in the foundation of the country. Malaysian policies and social distancing measures are based on daily infections and R0 (average number of infections per infected person), estimated from the data. Models of the Malaysian COVID-19 spread are mostly based on the established SIR compartmental model and its variants. These models usually assume homogeneity and symmetrical full mixing in the population; thus, they are unable to capture super-spreading events which naturally occur due to heterogeneity. Moreover, studies have shown that when heterogeneity is present, R0 may be very different and even possibly misleading. The underlying spreading network is a crucial element, as it introduces heterogeneity for a more representative and realistic model of the spread through specific populations. Heterogeneity introduces more complexities in the modelling due to its asymmetrical nature of infection compared to the relatively symmetrical SIR compartmental model. This leads to a different way of calculating R0 and defining super-spreaders. Quantifying a super-spreader individual is related to the idea of importance in a network. The definition of a super-spreading individual depends on how super-spreading is defined. Even when the spreading is defined, it may not be clear that a single centrality always correlates with super-spreading, since centralities are network dependent. We proposed using a measure of super-spreading directly related to R0 and that will give a measure of 'spreading' regardless of the underlying network. We captured the vulnerability for varying degrees of heterogeneity and initial conditions by defining a measure to quantify the chances of epidemic spread in the simulations. We simulated the SIR spread on a real Malaysian network to illustrate the effects of this measure and heterogeneity on the number of infections. We also simulated super-spreading events (based on our definition) within the bounds of heterogeneity to demonstrate the effectiveness of the newly defined measure. We found that heterogeneity serves as a natural curve-flattening mechanism; therefore, the number of infections and R0 may be lower than expected. This may lead to a false sense of security, especially since heterogeneity makes the population vulnerable to super-spreading events.

**Keywords:** COVID-19; network heterogeneity; R0; super-spreaders; SIR on networks; contact networks; weighted networks; Malaysia

## 1. Introduction

COVID-19 shook the world and made us very aware of the number of infections, the growth of which must be flattened to ensure health services are not overwhelmed. This curve, representing the number of infections, is commonly modelled by the Susceptible-Infected-Recovered (SIR) model [1] and its variants. The basic SIR compartmental model assumes that the population is compartmentalized into three compartments, namely the 'Susceptible', the 'Infected' and the 'Recovered'. In its simplest form, the infection between these compartments will be symmetrical in nature, with a constant infection rate $\beta$ governing the 'Susceptible' and 'Infected' compartments and the constant recovery rate $\gamma$ gamma governing the interactions between 'Infected' and 'Recovered' [2].

The underlying assumption of a typical SIR compartmental model is that communities are homogeneous, such that the population is composed of individuals who mix uniformly and randomly infect each other [1]. A more representative and realistic model of epidemic spread needs to incorporate heterogeneity, particularly for a multi-ethnic and diverse country like Malaysia. Heterogeneity in epidemic modelling can be added in many ways [3], such as varying individual parameters (ethnicity, age, gender, contact rate and compliance to public health recommendations, as well as disease-dependent individual parameters), susceptibility to disease, transmission rate, mode of transmission and recovery rate. Efforts to incorporate heterogeneity through compartmental modelling usually focus on varying infection rates between compartments [4] and increasing the number of compartments [5]. A natural way to incorporate heterogeneity is by modelling the spread on a network.

A network or a graph can represent a set of individuals (vertices) connected with each other through relationships or physical contact (edges), visualized in Figure 1. In this article, we shall use the term graph and network interchangeably. The spread on a compartmental SIR model may be compared to an epidemic spread on a regular graph or a complete graph [3,6], which is symmetrical in nature. A network representing human contact is usually heterogeneous and asymmetrical in nature, since relationships and contact of individuals vary. This breaking of symmetry makes the calculation of transition probabilities and effective infection rates more complicated, compared to the compartmental SIR model. However, modelling the spread on networks enables and, in fact, predicts the existence of super-spreaders and clusters within the population.

There are methods to define super-spreading events [7] or incorporate information of super-spreading localities [8], but in this article, we shall focus on defining the super-spreading individuals (vertices) within a network. Quantifying a super-spreader is directly tied to the idea of importance and centrality of a vertex. The spreading capacity of a vertex can be measured by how much of an outbreak it can cause by being infected. This may be referred to as influence maximization [9]. The spreading capacity can also be measured by how much deleting a vertex would reduce the expected outbreak size, as we previously investigated [10], in relation to sentinel surveillance [9] and strategizing to minimize infections. These capacities may be captured by measures such as degree and centralities, depending on the structure of the underlying network. Measures such as vitality [9] and core periphery structure [11] are also related to various centralities. The authors of [8] defined a super-spreading measure that incorporated R0 into its calculation, thus defining a centrality weighted by R0 for each locality. Therefore, the definition of super-spreading individuals (vertices) is highly dependent on how super-spreading is defined. Even when the spreading is defined, it may not be clear that a single centrality always correlates with super-spreading [9,10] due to the heterogeneity of the network. We propose using a measure of super-spreading directly related to R0, a measure well-established in the epidemic spreading literature. This measure will correlate to 'spreading' in the R0 sense, regardless of the underlying network.

R0 (R-naught) is the average number of infections per infected person. When R0 is greater than one, infections can spread in a totally susceptible population [12]. For a successful eradication of the virus, it is vital that R0 is forced to go below one, so that the virus eventually dies out. Therefore, the value of R0 serves to quantify the 'spreading' potential of the virus. R0 is used by governing bodies to plan for future preventive actions. The Malaysian government in particular even had a special press conference to explain R0 and how it will affect future Malaysian policies in light of COVID-19. In the basic SIR model, R0 can be calculated by taking the ratio of infection rate per recovery rate, such that $R0 = \frac{\beta}{\gamma}$ [6,13]. However, the asymmetrical and heterogeneous structure of a network means that calculating R0 will not be as straightforward [12,13].

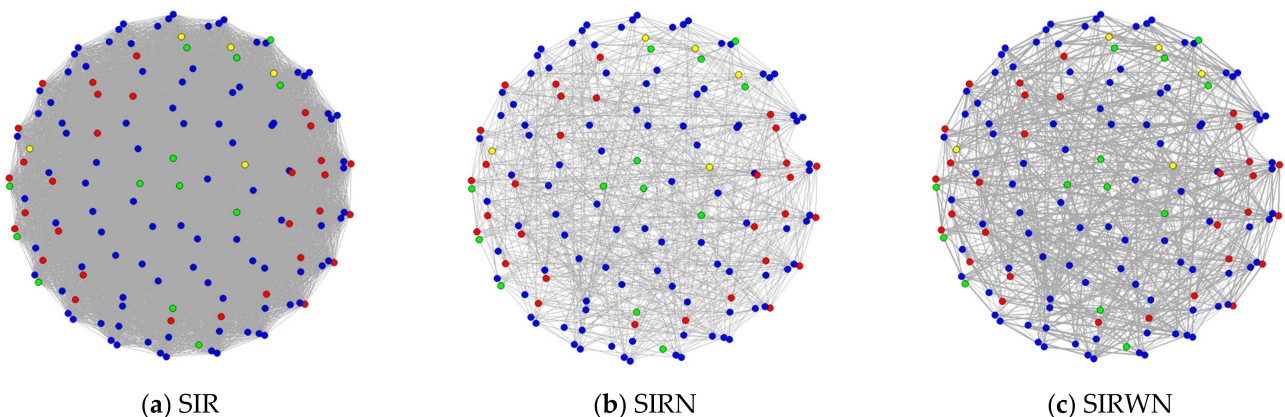

**Figure 1.** A network of 148 individuals as vertices linked by edges of potential infections. Colors represent different ethnicities of these individuals. In (**a**), homogeneity is assumed. All individuals are connected and can randomly infect anyone. In (**b**), an individual can only infect their friends as stated in a questionnaire [14]. In (**c**), an individual can only infect their friends in [14], but the chances of infecting is higher for a close friend. A close friend is indicated by a thick edge.

We simulated the SIR-like spread on the network using a Markov Chain Monte Carlo process, repeated for 1000 iterations. Thus, we defined a novel measure of spreading, $\alpha_S$, by taking the ratio of iterations, where R0 grows larger than 1 over all iterations. $\alpha_S$ represents the fraction of the simulation in which the epidemics spread. The larger this value is, the higher the chances of epidemic spread. We demonstrate that this value is sensitive to network heterogeneity as well as heterogeneity of effective infection rates.

Therefore, the outline of this article is as follows. In Section 2, we discuss epidemic modelling and illustrate the Malaysian COVID-19 situation. We estimated the daily infection and recovery rate from recent data, including data from various clusters of infections. In Section 3, we describe how Malaysian heterogeneity can be captured through networks, and we outline an algorithm to simulate SIR-like spread on networks in Section 4. We modelled the spread of COVID-19 on a real Malaysian network of university students with varying levels of heterogeneity. The effect of heterogeneity was quantified using three main measures: (1) the fraction of infected individuals in the population, (2) R0 (average number of infections per infected person) and (3) the chances of epidemic spread, $\alpha_S$ (fraction of cases where R0 grows larger than 1). Furthermore, in Section 5, we simulate the dynamics of super-spreading events within this heterogeneous network and, consequently, highlight the differences in the outcome. To conclude we will discuss the implications of our findings for the literature of the quantification of super-spreading in general and, specifically, to the modelling of the Malaysian epidemic spread.

## 2. The Spread of COVID-19

To understand COVID-19, many different models have been proposed for populations in distinct countries [15–18]. The Global Epidemic and Mobility Model (GLEAM), a meta-population-based, spatial epidemic model, has utilized transport networks to project the impact of travel limitations on the national and international spread of the epidemic. GLEAM [16] highlighted that the travel quarantine of Wuhan delayed spread at the international scale, where case imports were reduced by nearly 80% until mid-February. The estimation of contact networks in the population is the underlying pillar of GLEAM [2].

Such computationally intensive data-driven simulations require inputs of heterogeneity relating to the contact of the underlying population through households, workplaces and schools, as well as transportation between these places, to construct estimations of contact networks. Modelling at an individual level implies trying to estimate the interaction of every single individual in the population. These models have been known to give more precise epidemic predictions for not just the United Kingdom, United States and

European Countries [15,16], but also for our neighboring countries, like Singapore [17] and Thailand [19].

Its small size and dominance of public transport has led to Singapore being more successful in incorporating data into its modelling [17,20]. At the very least, Malaysia has data of the population, collected through census and surveys at the Department of Statistics Malaysia (DOSM), as well as contact tracing data at the Malaysian Ministry of Health (MOH) [21]. To our knowledge, modelling that integrates both these datasets does not yet exist for Malaysia. Privacy concerns aside, we believe that integration of these datasets will be highly beneficial for the modelling of COVID-19 in Malaysia, especially due to the heterogeneous nature of the population.

DOSM [22] estimates that the 32.7 million population of Malaysia consists of 29.7 million citizens and 3 million non-citizens. The configuration by gender is almost equal between males and females. Partitioning by ethnicity, the Malays and Bumiputeras are the majority, with 69.3%, followed by Chinese (22.8%), Indian (6.9%) and others with 1%. The structure of Malaysian citizens by age is largely dominated by the working population aged 15 to 64 (69.7%), with 0–14 years (23.5%) and 7% of populations aged 65 and older. These statistics indicate that Malaysia is a country ripe with its own unique heterogeneities.

### 2.1. The Spread of COVID-19 in Malaysia

COVID-19 was first recorded in China back in November 2019. It was declared a pandemic by the World Health Organization (WHO) in 2020. In Malaysia, the first case was recorded on the 25 January 2020 by three Chinese nationals who previously had close contact with an infected person in Singapore [5]. The significant impact of a Tabligh gathering in Sri Petaling on 27 February to 1 March 2020 was widely viewed as a main factor influencing the issuance of the Movement Control Order (MCO) by the Malaysian government.

The number of confirmed positive cases in Malaysia continues to increase, and it reached its peak in April 2020. When the number of active cases showed a declining trend, the lockdown restrictions were relaxed over the next several months to Conditional Movement Control Order (CMCO) and Recovery Movement Control Order (RMCO). However, since mid-September 2020, several waves of the virus have been hitting Malaysia, leading to multiple versions of lockdown. Currently, the whole country is back to the MCO restrictions, save for certain relaxation on economy activities.

The Malaysian Ministry of Health (MOH) is the main source of formal information related to the COVID-19 spread, via briefings by the Director General (DG) of Health and via the MOH website [21]. These daily statistics include the number of confirmed positive cases, number of active cases, number of recovered cases, number of deaths, number of cases in Intensive Care Unit (ICU) and number of intubated cases. These statistics are produced not only for Malaysia as whole, but also for localities, such as districts and states.

On 18 October 2020, there was a special press conference by the DG to explain R0 and how it would be crucial for determining the future MCOs. The MOH COVID-19 portal [21] explains that R0 is the average number of infections by one infected person and is thus indicative of the current spreading rate. The portal also provides daily R0 estimates for the whole country and various states. These values are then used by the government to determine and plan for future preventive actions. In this way, the number of infections and R0 directly affects Malaysian policies and restrictions.

### 2.2. Modelling the Malaysian COVID-19 Spread

The basic SIR compartmental model is governed by the equations

$$\frac{dS}{dt} = -\beta SI, \quad \frac{dI}{dt} = \beta SI - \gamma R, \quad \frac{dR}{dt} = \gamma R, \tag{1}$$

where $\beta$ is the infection rate, and $\gamma$ is the recovery rate [1,2]. This model assumes that the population is compartmentalized into three compartments, namely the 'Susceptible', the

'Infected' and the 'Recovered'. In (1), $S$ is the number of individuals being in the 'Susceptible' state, $I$ is the number of 'Infected' individuals and $R$ is the number of individuals in the 'Recovered' state. The underlying assumption is that communities are homogeneous, such that the population is comprised of individuals who mix uniformly and randomly infect each other [15]. This assumption is generally true for any SIR-type compartmental model, including the SEIR (the model used by the Malaysian MOH [21]).

Heterogeneities in the population and the interactions between individuals profoundly affect the dynamics of infections. One way to quantify the dynamics of an epidemic spread is using R0. R0 can be defined as the average number of secondary cases caused by an infectious individual in a completely susceptible population [13]. We use R0 here to avoid confusion with the 'Recovered' compartment. We acknowledge that R0 is usually also labelled $R_0$, R-Naught (MOH website [21]) or R in the literature. There are many ways to estimate the R0 of a model [23,24]. In a typical SIR model, R0 can be calculated from $R0 = \frac{\beta}{\gamma}$ [6,13] using $\beta$ and $\gamma$ are from Equation (1). This relationship is usually utilized to model the SIR of modelling of the Malaysian spread [10,18].

When R0 is greater than one, infections can spread in a totally susceptible population [13]. For a successful eradication of the virus, it is vital that R0 is forced to go below one, so that the virus dies out and cannot re-invade if immunity is maintained [13]. Salim [18] used the SIR model with different sizes of the $S$ (Susceptible) population to model different effects of the MCO. The R0 used in their study was the WHO estimate of 1.16. Wong [25] simulated the Malaysian SIR spread with various values of R0 to depict different strategies and vaccination program efficiency.

Alsayed [26] used an Artificial Neural Network (ANN) on data from 25 January 2020 to 5 April 2020 to estimate that the Malaysian infection rate is between 0.015 and 0.041. The estimates of 'Recovered' cases in [27] used data from 4 February 2020 to 16 May 2020, whereas in [28], data from 1 Dec 2020 to 31 January 2021 were utilized, resulting in an estimation of the transmission rate at 0.11 and the recovery rate at 0.026. Most models of the Malaysian COVID-19 spread assume homogeneity in modelling; we demonstrate that this could lead to inaccuracy in estimates of $\beta$ and $\gamma$.

*2.3. On the Estimation of Malaysian $\beta$ and $\gamma$*

We shall demonstrate, in this paper, the fact that using similar $\beta$ and $\gamma$ on the same population under various heterogeneity conditions will render a different number of infections and R0. Therefore, we estimated the $\beta$ and $\gamma$ from the most recent Malaysian data to be utilized in all of our simulations. MOH statistics [21] from 26 May to 8 June 2021 provided these variables: the number of tests, number of positive cases, number of recovered case and the number of active cases. Then, the daily infection rates and recovery rates were computed based on these formulas:

$$\text{Daily infection rate} \; = \frac{\text{Number of positive cases}}{\text{Number of tests}} \tag{2}$$

$$\text{Daily recovery rate} \; = \frac{\text{Number of recovered cases}}{\text{Number of active cases}} \tag{3}$$

Both these values were then averaged over the 14-day period. The summary statistics for these two parameters are given in Table 1. Referring to Table 1, the infection rates ranged between 0.5 and 0.1, with a mean of 0.0698. Similar figures were recorded for the recovery rate, with the mean being slightly higher than the infection rate at 0.0729.

We also took into account statistics reported for 28 ended infection clusters, as reported on the MOH website [21] on 9 April, 20 April and 21 April 2021. The number of positive cases and the number of tests conducted for each cluster were reported; hence, we used the same formula as in (2) to compute the infection rate. Details are given in Table 2. It should be pointed out that the values in Table 2 are larger than the ones in Table 1, as data in Table 2 are based on infection clusters, whereas Table 1 is for the whole country. Some of the clusters were from construction sites with a high effective infection rate due

to the nature of contact (highlighting that indeed heterogeneity is significant), as high as almost 40%. Hence, the mean of the infection rates was also relatively high in these clusters. We observed that the first quartile value was 0.12, which is close to the maximum daily infection rate, at around 0.08.

**Table 1.** The summary statistics of daily infection and recovery rates for 14 days (26 May–8 June 2021).

| Statistic | Infection Rate | Recovery Rate |
|---|---|---|
| Min | 0.0627 | 0.0592 |
| Maximum | 0.0808 | 0.0896 |
| Mean | 0.0698 | 0.0729 |
| Median | 0.0696 | 0.0709 |
| Standard deviation | 0.0051 | 0.0086 |
| First quartile, Q1 | 0.0657 | 0.0661 |
| Third quartile, Q3 | 0.0728 | 0.0799 |

**Table 2.** The summary statistics of infection rates for 28 ended clusters of infection.

| Statistic | Infection Rate |
|---|---|
| Min | 0.0600 |
| Maximum | 0.3897 |
| Mean | 0.2222 |
| Median | 0.2259 |
| Standard deviation | 0.1079 |
| First quartile, Q1 | 0.1249 |
| Third quartile, Q3 | 0.3185 |

The basic SIR model can be solved mathematically to obtain an exponential rate of infection and recovery [6,29]. In an exponential distribution, the rate of recovery is inversely proportional to the average recovery time. Therefore, if we take 0.0728 as our recovery rate, the average recovery time in the simulations will be 13.5 days. This is consistent with the information given by the Malaysian DG, Tan Sri Dr. Noor Hisham Abdullah in The Edge Markets [30], stating that it took a COVID-19 patient around 10 to 14 days to recover. Thus, one could say that the recovery rate is between 0.0714 and 0.1. This information is also supported by the daily figures of daily R0 reported by the MOH [21] for different states, which is based on the moving window of 7 and 14 days. Judging from all of this information, we proceeded with the analysis by setting the infection and recovery rates to be $\beta = 0.0698$ and $\gamma = 0.0729$, respectively.

## 3. Heterogeneity and R0

When heterogeneity is taken into account, R0 estimates from modelling will differ from homogeneous cases [12,13]. The accuracy of R0 prediction is important, since R0 is a utilized evaluation of effectiveness of country-specific public health intervention strategies [23], including Malaysia. Accurate estimation of R0 is crucial for predicting the herd immunity threshold needed to stop transmission. Shaw and Kennedy [24] claimed that there is tremendous uncertainty when attempting to use R0 for public health planning, since it alone cannot predict future dynamics.

Heterogeneity in epidemic modelling can be added in many ways [3], such as varying individual parameters (ethnicity, age, gender, contact rate and compliance to public health recommendations, as well as disease-dependent individual parameters), susceptibility to disease, transmission rate, mode of transmission and recovery rate. These parameters can be defined in a compartmental model using various specific rates and compartments [5], but arguably, the most natural and intuitive way to incorporate heterogeneity is by modelling the spread on a contact network.

### 3.1. Contact Networks

Due to COVID-19, contact tracing is now a household word in Malaysia. When the contacts of an infected person are traced, one can form a picture visualizing the spread of the epidemic. This picture illustrating human connections is known as contact networks [11,31]. Contact networks have been used to model previous outbreaks, such as 2019 H1N1 [2] and the MERS in 2015 [11]. The structure of contact networks can readily explain the existence of super-spreaders and clusters formation in epidemic spreading [11,31,32], since these behaviors are common observations in real networks, known in network analysis as hubs and communities, respectively [6]. GLEAM [16] modelled the COVID-19 spread at an international scale, by estimating the underlying contact networks.

Using contact networks adds heterogeneity to the modelling not only by exactly specifying who is in contact with whom, but through networks that can account for age, co-morbidity, gender, different types and strains of viruses or mutating pathogens that change infection rates [32]. Various population networks display a large variety of heterogeneity [29]. Networks are very useful to quantify the extent to which real populations depart from the homogeneous-mixing assumption directly affecting the resulting epidemiological dynamics. Malaysian interaction networks have been modelled in [14,33].

Network heterogeneity is a double-edged sword. On one hand, the communities formed mean that not everyone has a chance to spread the virus, with everyone thus delaying the spread and reducing the peak. On the other hand, the structure of the network also makes it vulnerable to infections of certain strategically positioned individuals, serving as bridges between communities. If these individuals are infected, the chances of spread are higher than the rest.

### 3.2. Malaysian Heterogeneity in University Friendship Networks

Network analysis is a tool based on graph theory. A graph can be written as $G = (V, E)$, where $V$ is the set of vertices, and $E$ is the set of edges [6]. Graphs are combinatorial objects used to model relations between elements of a system. The relations considered in our simulations are symmetrical in nature, since the vertices (contacts) can give each other the disease. Therefore, the set of edges, $E$, considered here is undirected, and the graph $G$ is an undirected graph.

Typically, a graph or network is depicted as a set of dots for the vertices, joined by lines or curves for the edges, as in Figure 1. In this article, we shall use the term graph and network interchangeably. We define the set of neighbors of a vertex as the vertices connected to it via an edge. Figure 1 is a network, with the set of vertices, $V$, representing 148 individuals with a varying set of edges, $E$.

In Figure 1a, we use the basic SIR assumption of homogeneity, where all individuals can be in contact with one another and can randomly infect any of the 148 individuals. The spread on a basic compartmental SIR model is generally equivalent to an epidemic spread on network in which the set of edges for each vertex will be all the other vertices [3], i.e., a complete network [6]. This means that there are 147 edges for each vertex. We shall refer to simulations done on this type of network as SIR simulations.

One way to quantify social interactions and contact networks is through friendship. This is especially true in a lecture hall setting, where students tend to be in close proximity and in contact with their friends. Therefore, we utilize a friendship network collected in a pre-Covid physical class setting through questionnaires [14,34]. Informed consent was obtained from all subjects involved in the study. The data sets were anonymized, stored and analyzed in a secure environment. In this questionnaire, the students identified who their friends were within the class and ranked their friendship from strongest to weakest. The maximum number of friends (edges) per student was 10, and the average number of friends (edges) was 9.

The class was comprised of Malaysian students of various ethnicities (represented by colours in Figure 1), genders and educational backgrounds. Table 3 summarizes the composition of the data samples based on the race factor. In Figure 1b, the set of edges, $E$,

representing the friendships amongst the 148 students, had 666 edges, and the potential spreading would be reduced since only friends could be infected. The spread would be heterogeneous due to the different number of friends of each individual. We shall refer to simulations done on this type of network as SIRN simulations.

**Table 3.** The distribution of the 148 students based on race.

| Race | Frequency | Percentage |
|---|---|---|
| Malay | 99 | 66.89 |
| Chinese | 31 | 20.95 |
| Indian | 13 | 8.78 |
| Others | 5 | 3.38 |
| Total | 148 | 100 |

In Figure 1c, the set of edges, *E*, representing the friendships amongst the 148 students, was similar to the one in Figure 1b, but now, the edges were weighted. Therefore, the network in Figure 1c is a weighted network. We obtained the weights on the edges from the student ranking in the questionnaire [14], and we categorized their friends into two categories of friends and close friends. The chances of an individual infecting a close friend would be higher than the chances of infecting a friend, thus adding more heterogeneity through varying infection rates. We shall refer to simulations done on this type of network as SIRWN simulations.

In this paper, we explicitly simulated two types of heterogeneity through networks. The first was heterogeneity in terms of the number of contacts in SIRN; we shall refer to this as network heterogeneity. The second was heterogeneity in terms of infection rate (more infectious for close friends) in addition to network heterogeneity in SIRWN. Furthermore, the edges in Figure 1b,c also reflect the underlying communities in the network and homophily (tendency for individuals to bond with people who are similar to themselves), thus illustrating some unique Malaysian heterogeneity.

## 4. SIR-like Simulations with Random Seed

There are various approaches to model the epidemic spread on networks [1,2,6], including agent-based approaches [15,16]. In this paper, we took a simple approach of simulating SIR-like behavior in the population using a modified algorithm from the one outlined in [10]. This simulation approach utilizes Monte Carlo simulations and is generally less computationally demanding compared to directly simulating differential equations in (1). As defined in [35] and [36], a Monte Carlo simulation is a set of iterative procedures with the inclusion of randomness to obtain numerical results.

The SIR process is probabilistic; thus several iterations are required to get a representative outcome, as the location of the initial infected vertex (seed) in the network will impact the speed at which the epidemics spreads through the system. Thus, the SIR, SIRN and SIRWN simulations were each simulated for 1000 iterations, and for each iteration, a random initial seed was chosen. To display a representative outcome of the 1000 iterations, the median value at each time step was plotted to portray the dynamics of infections and R0.

All simulations in this paper were simulated with the population size $N = 148$, infection rate $\beta = 0.0698$, recovery rate $\gamma = 0.0729$ and time steps (days) up to a maximum of 100 days. The spread in the population is unimpeded and dies out due to infection and recovery rates. One hundred days was chosen as the maximum, since more than 80% of our simulations had a fully recovered population by day 100. We assumed that the spread was fully contained in an isolated 'bubbled' setting, with no external influences or interactions. In a network setting, transmission can only occur between vertices that are connected to each other through an edge; we refer to such vertices as neighbors.

### 4.1. SIR, SIRN and SIRWN Simulations

The basic SIR model assumes homogeneity and full mixing, i.e., all individuals interact with all other individuals and with the same probability at all times. Thus, for the SIR simulations on the network in Figure 1a, set $E$ contained an edge between all pair of vertices. Since we defined the neighbors of a vertex as all the other vertices connected to it through an edge, the neighbors of each vertex were the rest of the population in the SIR simulations.

Starting with one 'Infected' vertex chosen at random as the seed, the main mechanism of the SIR simulations is summarized in these steps:

1. 'Infected' vertices are set to recover exponentially at rate $\gamma$. 'Recovered' vertices are not infectious and can no longer be infected;
2. 'Infected' vertices may infect all 'Susceptible' neighbors at an exponential rate $\beta$ at each time step, until they recover;
3. Newly 'Infected' neighbors join the other 'Infected' vertices, and repeat 1 and 2 until there are no more neighbors to infect.

Thus, at each time step, every vertex is in one of the $S$, $I$ or $R$ compartments. To implement step 2, we used a Metropolis Hastings Algorithm to sample from the exponential distribution with rate $\beta$. Details pertaining this algorithm can be found in [37].

For the SIRN simulations on the network in Figure 1b, the simulations steps were exactly the same as the SIR simulations, except that the set of edges $E$ now contained friendship as declared in the questionnaire [14]. Therefore, the neighbors of each vertex were their proclaimed friends, and all friends could be equally infected with rate $\beta$.

For the SIRWN simulations, the set of edges $E$ was similar to that in the SIRN simulations. However, not all neighbors were equal. A total of 63.8% of the SIRWN edges were close friends, and the rest were friends. The effective infection rate was varied, being higher for a close friend than a friend. Thus, the probability to infect close friends was made to be higher through alterations of the Metropolis Hastings Algorithm, so that the steps became:

1. 'Infected' vertices are set to recover exponentially at rate $\gamma$. 'Recovered' vertices are not infectious and can no longer be infected;
2. 'Infected' vertices may infect all 'Susceptible' neighbors at each time step until they recover. However, there are two types of neighbors, close friends and friends. Close friends are infected at an exponential rate slightly higher than $\beta$, while friends are infected at an exponential rate slightly lower than $\beta$;
3. Newly 'Infected' neighbors join the other 'Infected' vertices, and repeats 1 and 2 until there are no more neighbors to infect.

To implement step 2, we used a novel Metropolis Hastings Algorithm designed to simulate epidemic spread on weighted networks, resulting in higher infection rates for weightier (thicker) edges.

### 4.2. Infections and R0

Three main measures that we intend to highlight from the simulations are related to the infection, R0 and chances of spreading:

$$I(t) = \frac{\text{Number of 'Infected' at time } t}{N} = \text{fraction of 'Infected' in the population} \quad (4)$$

$$R0(t) = \frac{\text{Number of total infections in the population by time t}}{N - \text{Number of 'Susceptible' at time } t}$$
$$= \text{average number of infections by one 'Infected' by time t} \quad (5)$$

$$\alpha_S = \frac{\text{Number of iterations where } R0(t) \geq 1 \text{ for any } t \text{ in simulation type } S}{\text{Total number of iterations for simulation type } S}$$
$$= \text{fraction of the simulation in which the virus spreads} \quad (6)$$

Important thresholds to be observed are maximum $I(t)$ and $R0(t)$, the time when $I(t) = 0$ and $R0(t) = 0$, as well as the time after max $R0(t)$ when $R0(t) < 1$. Figure 2 visualizes Equations (4)–(6) for the 1000 iterations of the SIR simulations on the network in Figure 1a. The blue line is the median value calculated at every time step, and we took this median to be representative of the overall evolution of the simulation. However, there were 'good cases' in the simulations, in which $R0(t) < 1$ for every $t$ ], such that the infection does not spread, and the population is not completely infected, as visualized by the flattened grey curves in Figure 2. The value $\alpha_{SIR} = 0.955$, displayed in Figure 2a, indicates that 955 out 1000 iterations of SIR spread, but there were 45 simulations that were 'good cases', in which the population was not completely infected. This value captures the chances of spreading for each simulation type, and for SIR simulations, the chance of spreading is 95.5%.

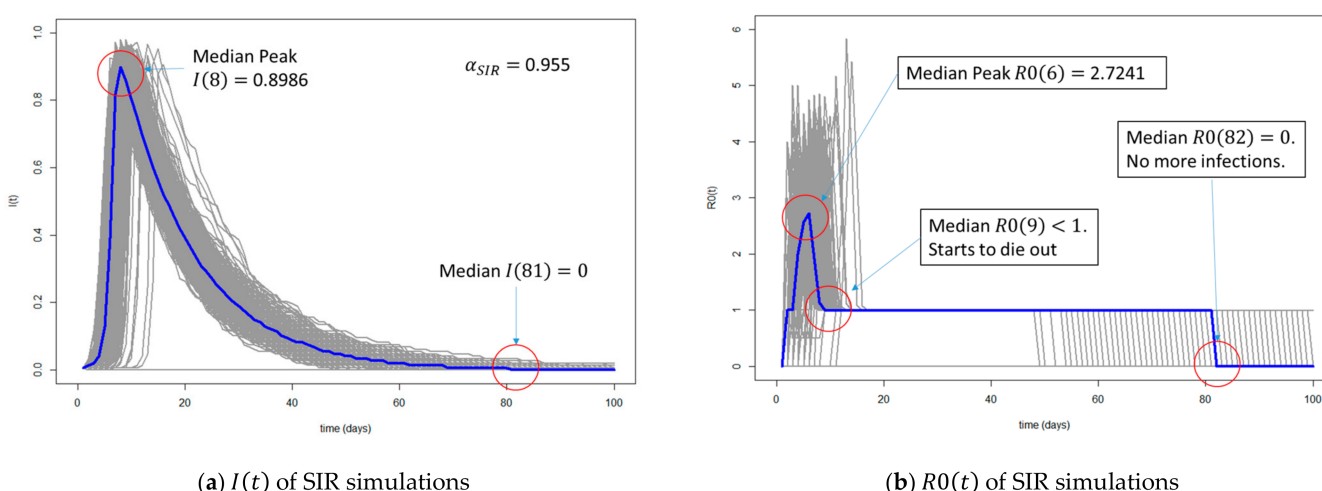

(**a**) $I(t)$ of SIR simulations        (**b**) $R0(t)$ of SIR simulations

**Figure 2.** Outcome of 1000 SIR simulations. The medians of the 1000 simulations are highlighted in blue. (**a**) $I(t)$ as defined in (4); (**b**) $R0(t)$ as defined in (5). An interesting threshold to be observed is the median peak at time 8, such that $I(8) = 0.8986$, and at time 6, such that $R0(6) = 2.7241$. Median $I(t)$ and $R0(t)$ went to 0 at time 81 and 82, respectively. The first time after the peak when $R0(t) < 1$ was at time 9 when the epidemic started to die out. Therefore, generally, for the SIR simulations, after 9 days, the infection starts to die out.

Figure 2a is a plot of $I(t)$, the dreaded infection curve that everyone wants to flatten; therefore, important values from the median curve are the maximum at day 8 of 0.8986 and the fact that it went to zero on day 81. This implies that on day 8, 89.86% of the population was infected, and the population was completely cured by day 81. Figure 2b is a plot of $R0(t)$, a number that in real life would have to be estimated from data or perceived infection rates. Because we had complete control of the spread, we were able to actually calculate R0 of the simulations, and from the median curve, the maximum of 2.7241 was at day 6 and it went to zero on day 82. This implies that at its peak, one individual infected almost three others, on average. By day 9, R0 went to below 1, thus indicating the start towards population recovery. Note, also, that for some simulations, R0 went up to more than 5 in Figure 2b, indicating that for some simulations, an 'Infected' may be infecting five others on average.

### 4.3. SIR, SIRN and SIRWN: Infections and R0

Similarly, we sought to highlight the measures in (4)–(6) for the SIRN and SIRWN. In Figure 3, we plotted the median $I(t)$ and $R0(t)$ of 1000 simulations for each type. Note that the blue curve in Figure 2 is now the black curve in Figure 3, and the measures highlighted in Figure 2 are now in columns headed by the black SIR heading in Figure 3. SIR, SIRN and SIRWN are in order of increasing heterogeneity. From Figure 3, we can say that heterogeneity serves as a natural curve-flattening mechanism. This is in line

with the observations of various other researchers that highlighted that R0 is reduced in heterogeneous population [24]. In fact, Ke et al. [23] suggest that the herd immunity threshold may also be lower. Diversity could indeed be a strength. However, this relatively late increase in infections may create a false sense of security in heterogeneous societies. Even though the chances of spreading are lower, as indicated by $\alpha_S$, the spread is still at least 90% of the simulations. More dangerously, this may lead to inaccurate estimates of infection and recovery rates.

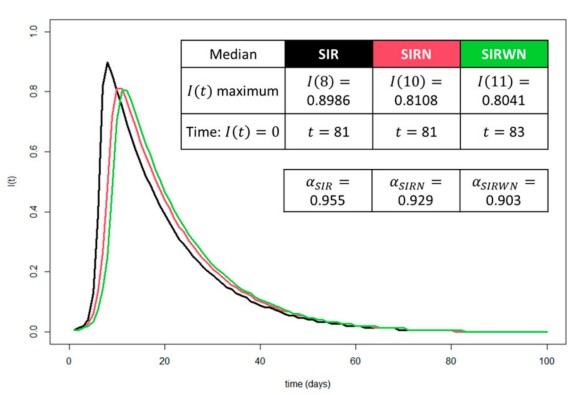

(**a**) Median $I(t)$ of SIR, SIRN and SIRWN

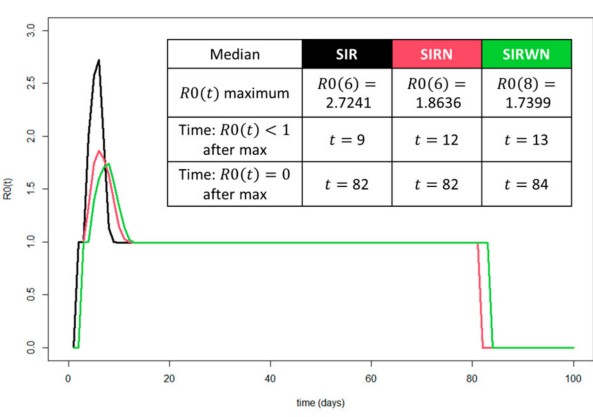

(**b**) Median $R0(t)$ of SIR, SIRN and SIRWN

**Figure 3.** Median $I(t)$ and $R0(t)$ of 1000 SIR, SIRN and SIRWN simulations colored black, red and green, respectively. (**a**) Median $I(t)$, as defined in (4); (**b**) Median $R0(t)$, as defined in (5). Peak $I(t)$ and $R0(t)$ were generally decreasing and delayed in order of increasing heterogeneity in SIR, SIRN and SIRWN. Chances of spreading, given by $\alpha_S$, also decrease with increased heterogeneity.

Recall that $I(t)$ and $R0(t)$ for all simulations are given by the same infection rate $\beta$ and recovery rate $\gamma$. Therefore, in Figure 3, we demonstrated that different $I(t)$ and $R0(t)$ values will be obtained by different kinds of homogeneity assumptions in the modelling, with similar rates of infection and recovery. This also implies that given an observed infection and R0 from data, the underlying infection rate $\beta$ may be higher and/or the underlying recovery rate $\gamma$ may be lower than the ones implied by a homogenous model. We strongly recommend estimation of infection and recovery rate to be performed on fully contract-traced clusters of infections, with complete information of the underlying contact networks, so that some form of heterogeneity can be taken into account. Inaccurate estimation of infection and recovery rates may be detrimental to future planning of health policies.

## 5. SIR-like Simulations with Fixed Seed

The next question we addressed was who are the super-spreader individuals in the network? In a homogenous population, everyone is equally connected to everyone, and this question has no answer, but in a heterogeneous population, there are a few possibilities. Using the heterogeneous structure of a network, it is possible to quantify structural differences between vertices. One way to quantify the importance of a vertex is using the concept of centrality which originates from the discipline of social network analysis [38]. Centrality gives vertices a rank.

Consequently, the next natural question would be does the ranking indicate the importance of the vertex, such that when it is infected, it will spread the virus faster than the rest? We have tried to address a related question in a previous study [10], where we demonstrated that by monitoring individuals with the highest Betweenness Centrality in the network and quarantining them as soon as they get infected, infections can be significantly reduced. This monitoring significantly flattened the infection curve when compared to random monitoring and monitoring based on a few other prominent network

measures. Therefore, in this paper, we utilized Betweenness Centrality to simulate super spreading dynamics.

### 5.1. Betweenness Centrality

A path is defined as a sequence of vertices, such that every consecutive pair of vertices in the sequence is connected by an edge. The distance $d_{ij}$ between two vertices, $i, j \in V$, is the number of edges along the shortest path. Every pair of vertices directly connected by an edge is at a distance of 1. The Betweenness Centrality (BC) of vertex $i \in V$ can be defined [6,38] as

$$x_i = \sum_{s \neq i, s \in V} \sum_{t \neq i,\ t \in V} \frac{n_{st}(i)}{N_{st}} \tag{7}$$

where $n_{st}(i)$ is the number of shortest paths from vertex $s$ to vertex $t$ that passes through vertex $i$. $N_{st}$ is the total number of shortest paths from $s$ to $t$. The more "in-between" other vertices a vertex is, the more central it is. High BC vertices act as connectors between different clusters formed in the network.

The highest ranked BC vertex in our network is colored red in Figure 4, and the lowest ranked is colored yellow. The red vertex represents the 'connector' individuals that are friends with many different communities in the network. On the other hand, the yellow vertex is only friends with one other individual in the network. Therefore, if an infection starts from the yellow vertex, whether the virus spreads or not, depends on this one edge. In order to rank by BC, we first calculated the BC of each vertices using (7) then ranked them from largest to smallest. The vertex with the highest BC was clear, but there were eight vertices with $x_i = 0$. In this case, we chose the yellow vertex from amongst the eight vertices, since it also has the lowest number of edges connected to it. Note that the red vertex does not have the greatest number of edges connected to it, despite having the highest BC.

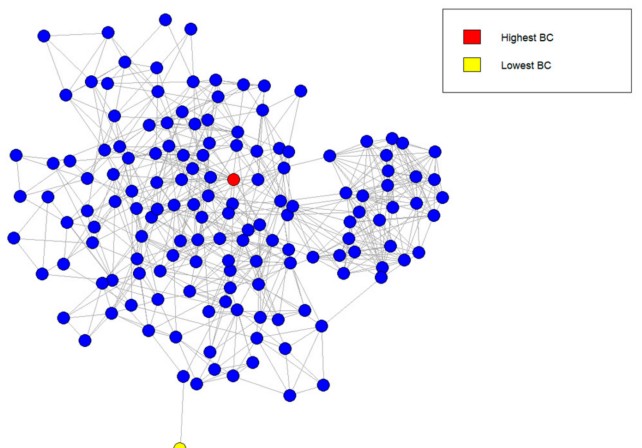

**Figure 4.** The network underlying SIRN and SIRWN, with vertices of the highest BC in red and the lowest BC in yellow. This is the exact same network in Figure 1b with different colorings.

### 5.2. SIRNBCH, SIRNBCL, SIRWNBCH and SIRWNBCL

The idea is that by choosing the vertex of the highest BC as the seed, one can simulate the worst-case scenario for the virus spread on SIRN- and SIRWN-type simulations, as this vertex acts as a bridge between communities and, thus, functions as a super-spreader. We named these super-spreading simulations with a fixed seed of the highest BC as SIRNBCH and SIRWNBCH, respectively. The algorithm for SIRN and SIRWN was utilized to simulate SIRNBCH and SIRWNBCH, except that the seed was no longer randomly selected for each of the 1000 simulations.

The highest BC vertex, colored red in Figure 4, was the fixed seed. Similarly, by fixing the seed to be the yellow vertex (lowest BC) in Figure 4, we simulated the best-case

scenario for the spread in SIRN and SIRWN simulations, named SIRNBCL and SIRWNBCL, respectively. The same steps from the algorithm for SIRN and SIRWN was utilized to simulate SIRNBCL and SIRWNBCL, except that the seed was fixed to be the yellow vertex. The results of the simulations are listed in Table 4. For comparison purposes, we also provided the results of Figure 3 in Table 5.

**Table 4.** $I(t)$, R0(t) and $\alpha_S$ of simulations with fixed seed.

| Median | SIRNBCH | SIRNBCL | SIRWNBCH | SIRWNBCL |
|---|---|---|---|---|
| $I(t)$ maximum | $I(9) = 0.8243$ | $I(1) = 0.0068$ | $I(11) = 0.8311$ | $I(18) = 0.4831$ |
| Time: $I(t) = 0$ | $t = 82$ | $t = 29$ | $t = 83$ | $t = 72$ |
| $R0(t)$ maximum | $R0(5) = 2.1429$ | $R0(1) = 0$ | $R0(7) = 1.8571$ | $R0(9) = 0.9932$ |
| Time: $R0(t) < 1$ after max | $t = 11$ | $t = 0$ | $t = 13$ | $t = 0$ |
| Time: $R0(t) = 0$ after max | $t = 83$ | $t = 51$ | $t = 84$ | $t = 73$ |
| $\alpha_S$ for 1000 simulations | $\alpha_{SIRNBCH} = 0.979$ | $\alpha_{SIRNBCL} = 0.528$ | $\alpha_{SIRWNBCH} = 0.959$ | $\alpha_{SIRWNBCL} = 0.615$ |

**Table 5.** $I(t)$, R0(t) and $\alpha_S$ of simulations with random seed plotted in Figure 3.

| Median | SIR | SIRN | SIRWN |
|---|---|---|---|
| $I(t)$ maximum | $I(8) = 0.8986$ | $I(10) = 0.8108$ | $I(11) = 0.8041$ |
| Time: $I(t) = 0$ | $t = 81$ | $t = 81$ | $t = 83$ |
| $R0(t)$ maximum | $R0(6) = 2.7241$ | $R0(6) = 1.8636$ | $R0(8) = 1.7399$ |
| Time: $R0(t) < 1$ after max | $t = 9$ | $t = 12$ | $t = 13$ |
| Time: $R0(t) = 0$ after max | $t = 82$ | $t = 82$ | $t = 84$ |
| $\alpha_S$ for 1000 simulations | $\alpha_{SIR} = 0.955$ | $\alpha_{SIRN} = 0.929$ | $\alpha_{SIRWN} = 0.903$ |

From Tables 4 and 5, one can see that $\alpha_{SIRNBCH} = 0.979$ and $\alpha_{SIRWNBCH} = 0.959$ were both larger than $\alpha_{SIR} = 0.955$, even though the infections and R0 of the SIR simulations were higher at $I(8) = 0.8986$ and $R0(6) = 2.7241$. Moreover, the R0 of SIRNBCH reached its peak at time 5, earlier than the peak of R0 in SIR at time 6. Thus, in SIRNBCH, the network heterogeneity lead to the increase in the chance of spreading and the hastening of peak R0. The network heterogeneity here was amplified by the targeting of the red vertex.

As for SIRWNBCH, we observed the conflicting effect of varying the infection rates on top of amplifying network heterogeneity. The varying infection rates resulted in SIRWNBCH having a lower $I(t)$, R0(t) and $\alpha_S$ compared to SIRNBCH, but it also led to SIRWNBCL having a higher $I(t)$, R0(t) and $\alpha_S$ compared to SIRNBCL. This can be explained by the fact that the yellow vertex only had one close friend. In SIRN, the infection rate is $\beta$ for any friend, but for SIRWN, the effective infection rate modelled by the simulation was higher than $\beta$ for a close friend; therefore, the chances of the infection spreading from yellow to her/his friend and the rest of population was higher in SIRWNBCL than in SIRNBCL. Heterogeneity of the infection rate cancelled out the extreme effects of the amplified network heterogeneity from the chosen fixed seed.

### 5.3. The Measuring of Super Spreading Events with $\alpha_S$

Recall from Equation (6) that $\alpha_S$ is defined as the fraction of the simulations that spreads when spreading behavior is quantified by $R0(t) > 1$. This value decreases as network heterogeneity increases, as observed from the SIR, SIRN and SIRWN comparison in Figure 3 and Table 5. It also decreases with an increased heterogeneity in infection rate, as observed from contrasting SIRN and SIRWN, as well as from SIRNBCH vs. SIRWNBCH. However, when SIRNBCL and SIRWNBCL were contrasted, $\alpha_S$ increased due to the focus on the single edge between yellow and his/her only friend. Therefore, $\alpha_S$ is sensitive to the structure of the network.

$\alpha_S$ is able to capture information not captured by the $I(t)$ and $R0(t)$. Recall that the population size is $N = 148$. We highlighted that the number of edges in the underlying network of SIRN in Figure 1a is actually 10,878 edges, since each vertex has 147 edges, representing a connection with all other vertices due to the homogeneity assumption. Conversely, the network underlying SIRN and SIRWN only has 666 vertices representing friendships. Therefore, the SIR has more than 16 times the edges of SIRN, and thus, the higher $I(t)$ and $R0(t)$ in Figure 3 is expected; even under these vastly different connectivity circumstances, we have shown that $\alpha_S$ of the SIRN and SIRWN may be larger than the SIR when a strategically positioned seed is chosen, as demonstrated through SIRNBCH and SIRWNBCH.

## 6. Discussion

The measure $\alpha_S$ is a relatively simple measure derived from R0 to quantify the chances of spreading and to be an indicator of potential super-spreading events. We are fully aware that to obtain knowledge of potential super-spreader individuals using BC requires full knowledge of the network, which may be difficult to obtain for large populations. However, the degree (the number of contacts) can be calculated more readily without even forming the network. Moreover, using contact-tracing data on clusters of infection, it may be possible to estimate and extrapolate towards forming a more complete network. Furthermore, in the Malaysian context, R0 estimated from daily data [21] within a certain confidence range can be analyzed further to incorporate the possibility of these super-spreading events, in addition to the focus on infections and average R0.

The results from SIRNBCH, SIRNBCL, SIRWNBCH and SIRWNBCL highlight the competing effect of the network structure and the varying infection rates. In larger networks, we hypothesize that the network effect from the structure will be more profound and possibly trump any kind of barrier provided by heterogeneity of infection rates. We suspect that this is what happened during the Tabligh gathering, where connections were amplified. We also noted that $\alpha_{SIRWNBCH} = 0.959$ is quite close to $\alpha_{SIR} = 0.955$, indicating that the chances of spreading in SIRWNBCH is close to that of SIR simulations. This suggests that results in the SIR simulations may be similar to those with competing effects of heterogeneity. This will be a subject of future investigations.

The potential underestimation of the infection rates, which is vital for predictions and epidemic modelling, is more profound for populations with high levels of heterogeneity, such as Malaysia. The results in Figure 3 highlight that similar infection and recovery rates give different $I(t)$, R0(t) and $\alpha_S$ for varying heterogeneity levels. Depending on a prediction of a homogenous model incorporating infections and R0 obtained from data, this may lead one to believe that the infection rate is lower, thus leading to inaction. Therefore, our future direction for further understanding of Malaysian heterogeneities through networks will include analyzing the data from clusters of infection and the census from the Department of Statistics Malaysia [22].

Our network of $N = 148$ was relatively small. We note that while we expect our results to be generalizable to larger networks, the drastic effect of the Betweenness Centrality (BC) targeting could be the effect of the size of the network and its particular structure. Nevertheless, in addition to being a good tool for highlighting heterogeneity, it is also a realistic number of individuals to be kept in a working 'bubble'. Monitoring connections and contact within this 'bubble' is a good strategy to safely reopen [10] certain parts of the economy. A bubble approach has already been implemented by some industries and institutions, where teams within departments only attend work on certain prescribed days or specific hours. We hope to motivate the analysis of information collected from these approaches using networks.

Especially in the initial stages of the pandemic, contact tracing has played a crucial role in the Malaysian response. The implementation of thorough questioning and isolating all possible contact, even before symptoms are manifested, can be effective. WE suspect that, the contact network obtained from the contact tracing information is the reason why

the strategy of contact tracing and isolation is highly effective in reducing the number of cases. Moreover, the natural heterogeneity of Malaysian communities provides a semi-protection from some infections. Unfortunately, the network effect of super-spreading events, connecting individuals from diverse communities and localities, is able to break the natural barrier created by the communities. Social distancing through MCO can be viewed as an effort to break the chain of infection through the network.

Malaysia has a unique social interaction pattern. Moreover, social interactions do vary from country to country, as it is largely influenced by the culture and tradition for the specific country, such as bowing in Japan and kissing on the cheek in Turkey. Therefore, incorporating the social interactions and heterogeneity when modelling the spread of COVID-19 is crucial to capturing human-to-human interaction. The construction of data-driven population-based contact networks will enable the quantification of various traits in the Malaysian spread of diseases, as well as evaluation of past intervention strategies, like the MCO, and inform future strategies and their consequences. This knowledge will empower the authorities to make evidence-based decisions related to disease mitigation and suppression strategies, as well as vaccinations and surveillance of future threats. This would all be in an effort to reduce the pressure on the health system capacity and enable a sustainable approach to the re-opening of the economy in the future.

## 7. Conclusions

In this paper, we highlighted that quantifying heterogeneity and super-spreaders is very important to predict a more realistic epidemic spread. Consequently, we defined a unifying measure of $\alpha_S$ that quantifies spreading (thus super-spreading) using the well-established and widely adopted concept of R0. We simulated the spread of COVID-19 with a varying degree of homogeneity on a real Malaysian contact network. Infection and recovery rates were estimated from recent data. In the first part, we simulated the spread of the virus in the event of just one random individual being infected. We simulated homogenous spreads (SIR), spreads on the network (SIRN) and spreads on the weighted network (SIRWN). The three basic measures taken into consideration, $I(t)$, R0(t) and $\alpha_S$, highlighted that heterogeneity serves as a natural curve-flattening mechanism, and thus, it may lead to an underestimation of infection rate and a false sense of security. For the second part, we simulated the spread of one strategically chosen individual being infected to highlight the worst-case scenario and the best-case scenario for SIRN and SIRWN. The worst-case scenario was simulated by infecting the individual with the highest Betweenness Centrality (SIRNBCH and SIRWNBCH), and the best-case scenario was simulated by infecting the individual with the lowest Betweenness Centrality (SIRNBCL and SIRWNBCL). The results in Tables 4 and 5 highlight that for the worst-case scenario simulations, whilst $I(t)$ and R0(t) were lower than SIR simulations, the chances of spreading, as quantified by $\alpha_S$, were higher, highlighting that $\alpha_S$ is a good measure of a super-spreading event. The median plot of SIRNBCL was almost completely flat, with $\alpha_{SIRNBCL} = 0.528$. SIRWNBCH and SIRWNBCL were both less extreme when compared to SIRNBCH and SIRNBCL due to an extra element of heterogeneity affecting the infection rates. In a nutshell, generally, heterogeneity serves as a natural barrier against the spread of the virus, but it also allows for super spreading to occur when central individuals are infected; thus, policies should also take into account probabilities of super-spreading events, in addition to observing the number of infections and R0. We particularly want to motivate this in the Malaysian context, so that the population data collected through the census and the contact tracing data collected during COVID-19 will be integrated into a more heterogeneous model. This model can hopefully serve as a toolkit for public health decision-makers to utilize lessons from the past, make decisions in the present and plan surveillance, as well as intervention, for the future.

**Author Contributions:** Conceptualization, F.A.R.; methodology, F.A.R. and Z.H.Z.; software, F.A.R. and Z.H.Z.; validation, F.A.R.; formal analysis, F.A.R.; investigation, F.A.R.; resources, F.A.R. and Z.H.Z.; data curation, F.A.R. and Z.H.Z.; writing—original draft preparation, F.A.R.; writing—review



and editing, F.A.R. and Z.H.Z.; visualization, F.A.R.; project administration, F.A.R.; funding acquisition, F.A.R. and Z.H.Z. All authors have read and agreed to the published version of the manuscript.

**Funding:** F.A.R. and Z.H.Z. receive support from the UKM grant GUP-2021-046 and GP-2019-K005224.

**Institutional Review Board Statement:** Not applicable.

**Informed Consent Statement:** Informed consent was obtained from all subjects involved in the study.

**Data Availability Statement:** Details of the real network underlying some of the simulations are not publicly available. The utilized algorithm was outlined in the article.

**Conflicts of Interest:** The funders had no role in the design of the study; in the collection, analyses or interpretation of data; in the writing of the manuscript or in the decision to publish the results.

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
