# Peer review of "Modelling Heterogeneity and Super Spreaders of the COVID-19 Spread through Malaysian Networks"

_symmetry, doi:10.3390/sym13101954_

Round 1

Reviewer 1 Report

Dear Editors,

the manuscript is consisted of sixteen pages, and it is organized in seven sections as follows. The analysis and conclusions are illustrated using four tables and four figures.

The first section gives the reader introductory facts about the significance of the considered problem, namely the authors’ motivation for choosing the problem. The study aims to simulate the spread of Covid-19 with varying degree of homogeneity on a real Malaysian contact network. The authors acquaint the readers with the used models in this work.

The next section is called “The spread of Covid-19 in Malaysia” and the authors give in more details the aim of the research, the current picture of the disease and there are identified the modelling of the Malaysian Covid-19 spread. The data is collected, processed, and analyzed appropriately. Moreover, the authors use most recent data and simulate homogenous spreads (SIR), spreads on the network (SIRN) and spreads on the weighted network (SIRWN). The authors consider different scenarios of infection.

In the next basic section (Heterogeneity and R0) the authors describe how Malaysian heterogeneity can be captured through networks and outline an algorithm to simulate SIR-like spread on networks in the next section “SIR like simulations with random seed”.

Furthermore, in section “SIR like simulations with fixed seed”, the authors simulate the dynamics of super spreading events within this heterogeneous network and they highlight the differences in the outcome.

The final two sections, called “Discussion” and “Conclusions”, summarize the results obtained and present detailed conclusions.  

At the end of the manuscript, the author presents a comprehensive list of 32 references.

The whole paper is well-organized and contains all of the components I would expect. The sections are well-developed and clearly explained. The paper is well-written and easy to understand. The authors need to check the numbering of the sections. The final two sections “Discussion” and “Conclusions” need to be numbered as 6 and 7, respectively.

In conclusion, the obtained results are meaningful, and they enrich its field of science, so I recommend the proposed paper to be accepted and published in Symmetry.

Author Response

We thank the reviewer for the nice comments and for pointing out our mistakes in numbering the discussion and conclusion. We have now corrected the numbering.

Reviewer 2 Report

In a nutshell, the presented work applies statistical techniques, measures and simulations to demonstrate that a heterogeneous population composed by different communities that tend to stick together has a different infection rate and infection distribution respect to an homogeneous population; therefore point out that measures based on homogeneous populations are not good representatives for heterogeneous population. The authors demonstrate this based on data collected for the Covid19 spread in the Malaysian population. They took as an example a set of 148 students of various ethnicities to simulate the different spreading that could take place. 

The overall paper is well written and well presented. However I have some doubts on the actual results usefulness. The article reads more like an exercise of all those statistical techniques and measures rather than an actual scientific contribution. The authors should better clarify the novelty of their approach especially in the introduction that is pretty poor and better explain how the obtained results are actually useful for the community.

Figure 1 presents coloured dots to show different ethnicities of students however it would be better to report in numbers the actual distribution, coloured dots are not so great especially for a color-blind person.

Many other countries present an heterogeneous population, how does the results obtained for Malaysian compare to other countries? Are there no studies for other countries?

Minor Comments:

-row 198: "is double edged sword" --> "is a double edged sword"

- row 494: "This suggest" --> "This suggests"

- row 525 "as an effort the break the chain" --> "as an effort to break the chain"

Author Response

(The response typed here is also uploaded as a pdf file)

Point 1: The overall paper is well written and well presented. However I have some doubts on the actual results usefulness. The article reads more like an exercise of all those statistical techniques and measures rather than an actual scientific contribution. The authors should better clarify the novelty of their approach especially in the introduction that is pretty poor and better explain how the obtained results are actually useful for the community.

Response 1: Thank you for reading the paper so thoroughly. We agree that the introduction should be better thus we have significantly extended the introduction and made changes the discussion, abstract and conclusion in line with this. We particularly focused on clarifying the novelty of our approach in defining super-spreading individuals directly in relation to R0 and highlighting how the results will be useful especially for the Malaysian community. Moreover sporadic changes throughout the paper have been made to accommodate for these additions and extra references were added. The introduction currently spans line 38 to line 118. Here we highlight line 70 to line 88:

“There are methods to define super-spreading events [7] or incorporate information of super-spreading localities [8] but in this article we shall focus on defining the super-spreading individuals (vertices) within a network. Quantifying a super-spreader is directly tied to the idea of importance and centrality of a vertex. The spreading capacity of a vertex can be measured by how much of an outbreak it can cause by being infected. This may be referred to as influence maximization [9]. The spreading capacity can also be measured by how much deleting a vertex would reduce the expected outbreak size as we previously investigated [10] in relation to sentinel surveillance [9] and strategizing to minimize infections. These capacities may be captured by measures such as degree and centralities depending on the structure of the underlying network. Measures such as vitality [9] and core periphery structure [11] are also related various centralities. [8] defined a super-spreading measure that incorporated R0 into its calculation thus defining a centrality weighted by R0 for each locality. Therefore the definition of super-spreading individuals (vertices) is highly dependent on how super-spreading is defined. Even when the spreading is defined, it may not be clear that a single centrality always correlates with super-spreading [9, 10] due to the heterogeneity of the network. We propose using a measure of super-spreading directly related to R0, a measure well-established in the epidemic spreading literature. This measure will correlate to ‘spreading’ in the R0 sense regardless of the underlying network.”

Point 2: Figure 1 presents coloured dots to show different ethnicities of students however it would be better to report in numbers the actual distribution, coloured dots are not so great especially for a color-blind person.

Response 2: We thank you for pointing this out. We have now added Table 3 to highlight the distribution based on race which is added right before Figure 1.

Point 3: Many other countries present an heterogeneous population, how does the results obtained for Malaysian compare to other countries? Are there no studies for other countries?

Response 3: Indeed there are many and we have added a portion in section 2 to illustrate the global situation and why were are keen to motivate more heterogeneous modelling in the Malaysian population. We highlight lines 121 to 144:

“To understand Covid-19, many different models have been proposed for populations in distinct countries [14, 15, 16, 17]. The Global Epidemic and Mobility Model (GLEAM), a meta-population based, spatial epidemic model have utilized transport networks to project the impact of travel limitations on the national and international spread of the epidemic. GLEAM [15] highlighted that the travel quarantine of Wuhan delayed spread at the international scale, where case importations were reduced by nearly 80% until mid-February. The estimation of contact networks in the population is the underlying pillar of GLEAM [2].

Such computationally intensive data-driven simulations require inputs of heterogeneity relating to the contact of the underlying population through households, workplaces, schools as well as transportation between these places to construct estimations of contact networks. Modelling at an individual level implies trying to estimate the interaction of every single individual in the population. These models have been known to give more precise epidemic predictions for not just the United Kingdom, United States and European Countries [14, 15] but also our neighbouring countries like Singapore [16] and Thailand [18].

Its small size and dominance of public transport has led to Singapore being more successful in incorporating data into its modelling [16, 19]. At the very least, Malaysia has data of the population collected through census and surveys at the Department of Statistics Malaysia (DOSM) as well as contact tracing data at the Malaysian Ministry of Health (MOH). To our knowledge modelling that integrates both these data sets do not yet exists for Malaysia. Privacy concerns aside, we believe that integration of these datasets will be highly beneficial for the modelling of Covid-19 in Malaysia especially due to the heterogeneous nature of the population. ”

Point 4: Minor Comments:

-row 198: "is double edged sword" --> "is a double edged sword"

- row 494: "This suggest" --> "This suggests"

- row 525 "as an effort the break the chain" --> "as an effort to break the chain"

Response 4: Thank you very much for pointing this out. We apologize for these careless mistakes. We have corrected them as suggested in lines 290, 593 and 624.

Round 2

Reviewer 2 Report

The authors addressed most of the comments I provided in the previous review, managing to provide a better Introduction section; they clarified some aspects of the experiment; they improved the discussion and conclusions sections.

This manuscript is a resubmission of an earlier submission. The following is a list of the peer review reports and author responses from that submission.